# The Impact of Highly Weathered Oil from the Most Extensive Oil Spill in Tropical Oceans (Brazil) on the Microbiome of the Coral *Mussismilia harttii*

**DOI:** 10.3390/microorganisms11081935

**Published:** 2023-07-29

**Authors:** Pedro Henrique F. Pereira, Luanny Fernandes, Hugo E. Jesus, Patricia G. Costa, Carlos H. F. Lacerda, Miguel Mies, Adalto Bianchini, Henrique F. Santos

**Affiliations:** 1Department of Marine Biology, Fluminense Federal University—UFF, St. Professor Marcos Waldemar de Freitas Reis, Niterói 24210-201, RJ, Brazil; phmicrobiol@yahoo.com.br (P.H.F.P.); luannyfernandess@gmail.com (L.F.); hugoemil@gmail.com (H.E.J.); 2Instituto de Ciências Biológicas, Universidade Federal do Rio Grande—FURG, Av. Itália, s/n, Carreiros, Rio Grande 96203-900, RS, Brazil; patcosta0@gmail.com (P.G.C.); adaltobianchini@furg.br (A.B.); 3Instituto Coral Vivo, Rua dos Coqueiros, 87, Santa Cruz Cabrália 45807-000, BA, Brazil; carloshenrique.lacerda@coralvivo.org.br (C.H.F.L.); miguel.mies@coralvivo.org.br (M.M.); 4Instituto Oceanográfico, Universidade de São Paulo, Praça do Oceanográfico, 191, São Paulo 05508-120, SP, Brazil

**Keywords:** bacteria, coral reef, 16S rRNA, 2019 Brazil oil spill, petroleum

## Abstract

In 2019, the largest oil spill ever recorded in tropical oceans in terms of extent occurred in Brazil. The oil from the spill was collected directly from the environment and used in an exposure experiment with the endangered reef-building coral *Mussismilia harttii*. The treatments of the experiment were control (without oil), 1% oil, 2.5% oil, and direct contact of coral with oil. The most abundant hydrocarbon in the seawater of the experiment was phenatrene, which is toxic to corals. However, overall, the concentration of PAHs was not very high. The analysis of the maximum photosynthetic capacity of Symbiodiniaceae dinoflagellates showed a small impact of oil on corals, mainly on the contact treatment. However, coral microbiomes were affected in all oil treatments, with the contact treatment showing the most pronounced impact. A greater number and abundance of stress-indicating and potentially pathogenic bacteria were found in all oil treatments. Finally, this highly weathered oil that had lain in the ocean for a long time was carrying potentially coral-pathogenic bacteria within the Vibrionaceae family and was able to transmit some of these bacteria to corals. Bacteria within Vibrionaceae are the main causes of disease in different species of corals and other marine organisms.

## 1. Introduction

Coral reefs are widely recognized for their remarkable biodiversity and high productivity, serving as key drivers in upholding essential ecological functions within the marine environment [1]. These important ecosystems provide numerous benefits, including supporting a wide array of marine species, acting as natural barriers against coastal erosion, and contributing to the health and resilience of the oceans [2]. Corals are crucial eukaryotic organisms that form intricate relationships with a wide range of microorganisms, encompassing both eukaryotic and prokaryotic species. These symbiotic associations involve Symbiodiniaceae dinoflagellates, fungi, endolithic algae, bacteria, archaea, and viruses, collectively referred to as the coral microbiome. These microorganisms play vital roles in maintaining the health and functioning of the coral host [3,4,5]. They contribute to a wide range of beneficial functions for the coral, such as the production and translocation of photosynthates, the supply of micronutrients, protection against pathogens, facilitation of nitrogen fixation, and defense against UV damage [6,7]. The coral animal, in partnership with its associated microorganisms, forms a dynamic and interconnected system referred to as the coral holobiont [8]. This holobiont experiences fluctuations in the composition and abundance of its microbial members, which are influenced by various factors, including environmental conditions and the daily requirements of the coral host [8].

Unfortunately, coral reefs are highly vulnerable to a range of anthropogenic activities, including the adverse effects of rising sea temperatures, ocean acidification, overfishing, and pollution, such as oil spills [9,10]. Most oil spills take place in the ocean, with an estimated annual volume of approximately 10^3^ tons [11]. Regrettably, the primary shipping routes for oil transportation traverse the oceans in proximity to vital coral reef sites, and some oil spill events in the past have occurred near coral reef areas [12]. Sadly, even in recent times, several incidents involving oil and its byproducts have transpired within coral reef regions, resulting in substantial coral cover losses [13].

The 2019 oil spill along the Brazilian coast was the most extensive spill recorded in tropical oceans worldwide and possibly the worst environmental disaster in South America. It covered a vast distance of approximately 2890 km and impacted 11 states of Brazil. The total volume of the spill was between 5000 m^3^ and 12,000 m^3^. The spill had a widespread impact on 57 coastal and marine protected areas (MPAs) along the Brazilian coast, surpassing the cumulative number of MPAs affected by all previous oil spills in Brazil’s history [14,15]. The origin of the oil spill remains unknown, making it an enigmatic orphan event in tropical oceans. Speculations suggest that the oil could be the result of illegal ship dumping or leakage from a historical shipwreck [15,16]. This immense volume of oil affected 10 important Brazilian ecosystems. The most affected ecosystems were estuarine water bodies (4929.74 km^2^ of area affected), followed by mangrove forests (489.83 km^2^), seagrass meadows (324.77 km^2^), beaches (185.3 km^2^), tidal flats (63.64 km^2^), intertidal hard bottoms such as sandstone reefs (45.95 km^2^), and shallow-water coral reefs (9.69 km^2^) [17,18]. However, little is known about the impact of this oil on marine micro- and macro-organisms, such as the microbiome and corals.

Changes in environmental conditions, such as contamination by oil, can lead to shifts in the composition and diversity of the microorganisms associated with corals, a phenomenon referred to as dysbiosis [19]. This disturbance in the microbial community can have harmful consequences for corals as it disrupts the intricate balance and functional diversity within the coral holobiont. Dysbiosis has been identified as a contributing factor to the heightened vulnerability of corals to diseases [20]. Nevertheless, the impact of oil contamination on the coral microbiome and its potential to trigger dysbiosis is not yet well comprehended.

Current knowledge regarding the effects of oil contamination on corals has identified the following effects: tissue death [21,22,23,24]; decreased growth rates [25,26,27,28]; changes in the Symbiodiniaceae dinoflagellates primary productivity [22,24,29,30]; excessive mucus production [23,31,32,33,34]; decreases in settlement and development of coral larvae [35,36,37,38,39]; and changes in the coral’s symbiotic microbiota [24,30]. Nonetheless, each coral reef ecosystem around the world is unique, including differences in their microbiomes. In addition to variations in the microbiome, differences in the physical-chemical characteristics of the environment and the composition of the oil can alter the impacts caused by the oil, resulting in different contamination outcomes that require investigation. Furthermore, the understanding of the impacts of weathered oil on coral microbiomes remains limited. Therefore, this study investigated the effects of weathered oil, resulting from the most extensive oil spill in tropical oceans, on the microbial communities of the reef-building coral *Mussismilia harttii*, an endangered and endemic species found in the Southwestern Atlantic.

It is crucial to conduct research focusing on tropical coastal ecosystems, which have received limited scientific attention and are relatively understudied, especially in the case of the South Atlantic reefs in Brazil, which are considered conservation priorities because of their higher resistance to climate change. These ecosystems play a significant role in the overall marine environment, and understanding their response to environmental impacts, such as oil spills, is essential for effective conservation and management strategies. By directing more research efforts towards these understudied areas, we can gain valuable insights into the unique challenges and vulnerabilities faced by tropical coastal ecosystems, thus enabling us to develop informed and targeted approaches to their preservation.

## 2. Materials and Methods

### 2.1. Experimental Design

The coral species evaluated in this study was *M. harttii*, a zooxanthellate scleractinian distributed along the northeast coast of Brazil and one of the main reef-building corals in the South Atlantic Ocean [40,41]. This species is endemic and listed as an endangered species along the Brazilian coast [42]. The oil used in the experiment was the same as the oil from the spill and was collected in situ before reaching the coast at Praia de Mucugê. This oil parcel was the first to be registered in the city of Porto Seguro, in the state of Bahia. The seawater and the corals used in the experiment were collected from the same site as the oil. The experiment was conducted for a total of 10 days at the Coral Vivo Research Station, Bahia, Brazil. Natural seawater from the oil collection site was used for all treatments, which were kept under the local irradiance (400 μmol m^−2^ s^−1^) and temperature conditions (≅26 °C).

The experimental setup encompassed 12 individual 8 L seawater tanks containing 3 polyps of *M. harttii*. The experimental treatments applied were (i) control (seawater only); (ii) 1% oil (80 g of oil added to seawater); (iii) 2.5% oil (200 g of oil added to seawater); and (iv) contact (80 g of oil placed on top of the polyps). Each treatment was applied in three replicate tanks. Before adding the oil to the tanks, it was first separated in 3 aliquots for DNA extracting.

### 2.2. Photosynthetic Efficiency

The impact of weathered oil on the photosynthetic capacity of the Symbiodiniaceae dinoflagellates associated with *M. harttii* was measured using pulse-amplitude-modulated (PAM) fluorometry (Walz GmbH, Effeltrich, Germany). The stress level imposed on the coral holobiont was determined by changes in the Fv/Fm ratio, which was obtained from dark-adapted samples. To measure the initial fluorescence signal (Fo), the modulated measuring light of the PAM (a weak pulsed light; <1 μmol photons m^−2^ s^−1^) was used, and the maximal fluorescence level (Fm) was estimated using a short saturating pulse of actinic light. The variable fluorescence (Fv) was calculated from Fm–Fo, and the maximum quantum efficiency of photosystem II (PSII) photochemistry was obtained from the ratio Fv/Fm. The diving-PAM system was configured with the following parameters: Measuring Light Intensity (MI) = 6; Saturation Pulse Intensity (SI) = 8; Saturation Pulse Width (SW) = 0.8, Gain (G) = 1; and Damping (D) = 1. We performed a Two-way ANOVA, followed by Tukey’s post-hoc test, using Systat 13 (Systat Software Inc., San Jose, CA, USA) to assess the significance of differences in the Fv/Fm ratio among treatments over time (treatments and time as independent variables).

### 2.3. Detection of Petroleum Hydrocarbons

The concentration of polycyclic aromatic hydrocarbons (PAH) in the water of each tank (control, 1%, 2.5%) was evaluated 4 and 10 days after the beginning of the experiment. PAH analysis was carried out using a gas chromatograph coupled with a mass spectrometer (Clarus 500—GC-MS; PerkinElmer, Waltham, MA, USA). An Elite-5MS silica capillary column (5% phenyl, 95% methylpolysiloxane; 30 m × 0.25 mm, 0.25 μm film thickness, PerkinElmer, Waltham, MA, USA) was used. The injector was kept at 280 °C in the spitless mode. The temperature started at 40 °C and was raised to 60 °C at an increasing rate of 10 °C/min. It was further raised to 290 °C at an increasing rate of 5 °C/min. Temperature was maintained at 290 °C for 5 min, then raised to 300 °C at an increasing rate of 10 °C/min and kept constant for 10 min. Helium was used as the carrier gas (1.5 mL/min). The MS operating conditions were interface 290 °C, ion source 200 °C, and electron energy 70 eV. Compound identification was based on individual mass spectra and GC retention times in comparison to literature, library data, and authentic standards. Standards were injected and analyzed under the same conditions employed for samples. Compound quantification was done using internal standards: 1-tetradecene for AHCs and naftaleno-D8, acenaftene-D10, fenantreno-D10, crisene-d12, and perileno-d12 for PAHs. Blanks were analyzed simultaneously with each batch of samples. Hydrocarbons identified and quantified included naphthalene, 1-methyl-naphthalene, 2-methyl-naphthalene, acenaphthylene, acenaphthene, fluorene, phenanthrene, anthracene, fluoranthene, pyrene and benzo[a]anthracene, chrysene, benzo[b]fluoranthene, benzo[k]fluoranthene, benzo[a]pyrene, indeno[1,2,3-cd]pyrene, dibenz[a,h]anthracene, and benzo[ghi]perylene.

### 2.4. DNA Extraction and Sequencing

To assess the bacterial and archaeal communities associated with the coral *M. harttii* after the oil contamination, at the end of the experiment, 1 polyp was collected from each tank (3 tanks per treatment) and macerated in a mortar under dry conditions, using a pestle. Total DNA was extracted from 0.5 g of the macerated tissue, using the PowerSoil^®^ DNA Isolation Kit (MOBIO, Qiagen—Hilden, Germany) following the manufacturer’s instructions. DNA was quantified using a Qubit fluorometer (ThermoFisher, Waltham, MA, USA) and subsequently stored at −80 °C.

High-throughput amplicon sequencing was performed with 2 × 300 nucleotides (nt) paired-end sequencing utilizing V3 chemistry. To target specific regions of the 16S rRNA gene in bacteria and archaea, the V4–V5 regions were amplified using the primer sets 515F (5′-GTGYCAGCMGCCGCGGTA-3′) and 909R (5′-CCCCGYCAATTCMTTTRAGT-3′) [43]. The sequencing was carried out on the Illumina Miseq platform (Illumina, San Diego, CA, USA) at the StarSeq GmbH company (Mainz, Germany).

### 2.5. Sequence Processing and Data Analysis

The raw sequence data was processed in the software QIIME2 2021.2 [44]. Using DADA2, the quality of reads was filtered [45], and amplicon sequence variants (ASVs) were aligned and classified against the Silva v.138 16S rRNA database [46]. Statistical analyses were performed in the R 4.0.4 environment [47]. Diversity and differential taxa abundance analyses were performed with the Phyloseq 1.34.0 package [48]. ASVs assigned to eukaryote, chloroplast, or mitochondria were removed, and a total of 620,311 reads, ranging from 27,632 to 55,896 reads per sample, were obtained. To minimize effects of sequence depth, prior to statistical analyses, samples were randomly normalized to an equal depth of 27,632 sequences per sample using the “rarefy_even_depth” function, leaving 4794 ASVs. To remove the individual effects between replicates, samples were merged into the treatment groups (control, 1%, 2.5%, contact, oil) using the “merge_samples” function. To assess structure similarity among the different treatments, Bray-Curtis distance matrices were used to create a multidimensional scaling ordination (MDS). Diversity significance was checked by Kruskal-Wallis and PERMANOVA tests, assuming *p*-values < 0.05. 

Prediction for potential biomarker or probiotic bacteria was reached by differential abundance analysis. Considering that differences in the taxa relative abundances are strictly linked to the treatment effect over the microbial community, we performed statistical tests to corroborate the findings. To do this, we used the mp_diff_analysis function from the MicrobiotaProcess package [49], using Kruskal-Wallis and Wilcoxon signed rank tests with 0.05 for the alpha value. The results were then displayed on a cladogram showing the hierarchical clustering of the samples based on the ASV differential abundances with the highlighted clades representing the differential species. The clade colors were related to each treatment, and the point sizes were scaled by the FRD (Kruskal-Wallis test). The final cladogram was generated by the mp_plot_diff_cladogram function [49].

## 3. Results

In order to assess the solubilization of PAHs in seawater, despite the prolonged presence of the oil, PAH concentrations were measured on the fourth and 10th days of the experiment. On the fourth day following the addition of weathered oil to the experimental tanks, the highest concentrations of PAHs were found for phenanthrene, fluorene, and anthracene, being 5.8 ng L^−1^, 1.09 ng L^−1^, and 1.64 ng L^−1^, respectively, in the presence of 1% oil. In the presence of 2.5% oil, these compounds were found to have concentrations of 1.37 ng L^−1^, 6.53 ng L^−1^, and 0.74 ng L^−1^, respectively (Figure 1).

After 10 days, phenanthrene, anthracene, pyrene, chrysene, benzo[a]anthracene, and fluorene also showed high concentrations in the presence of 1% oil; they were 4.48 ng L^−1^, 3.79 ng L^−1^, 2.35 ng L^−1^, 1.45 ng L^−1^, 1.31 ng L^−1^, and 1.12 ng L^−1^, respectively. In the presence of 2.5% oil, only phenanthrene, fluorene, and anthracene had concentrations >1 ng L^−1^, which were 6.32 ng L^−1^, 1.41 ng L^−1^, and 1.12 ng L^−1^, respectively. There were no substantial differences in the concentrations of solubilized PAHs in seawater when using 1% or 2.5% oil. The seawater collected at Mucugê Beach and utilized in the experiment revealed a low concentration of PAHs, which falls within normal limits.

Changes in the Fv/Fm ratio were employed as a metric to assess the stress level endured by the coral holobiont. In this regard, the impact of the weathered oil on the photosynthetic efficiency of the Symbiodiniaceae dinoflagellates associated with *M. harttii* exhibited a modest yet statistically significant reduction (*p* < 0.05) in the Fv/Fm ratio on days 2 and 3 in all treatments with oil (1% oil, 2.5% oil, and contact), compared with the control treatment (Figure 2). From day 4 to day 8, all treatments demonstrated similar Fv/Fm values. However, beginning on day 9, the Fv/Fm values in the treatment where the oil was in direct contact with the coral started to decrease.

Despite the Symbiodiniaceae dinoflagellates associated with *M. harttii* not displaying substantial changes in photochemical efficiency across all treatments, the MDS analysis shows some dissimilarity on the microbiome associated with the corals in the different treatments (Figure 3). The treatment where oil was in direct contact with the coral exhibited a smaller difference compared to the oil microbiome. This treatment, among all the ones that received oil, exhibited the largest divergence in comparison to the control treatment. It is also evident that in general, the oil possesses a distinct microbiota compared to corals.

The heatmap analysis revealed that the most abundant ASVs found in the weathered oil from the most extensive oil spill in tropical oceans were species within the Rhodobacteraceae family, *Vibrio* sp., *Alteromonas* sp. *Pseudoalteromonas* sp., and the Vibrionaceae family, *Thalossospira* sp. and *Thalassolituus* sp. (Figure 4). Among these ASVs, the taxa *Vibrio* sp., *Alteromonas* sp., Vibrionaceae family, and *Thalassospira* sp. were also found in corals from oil treatments and were not found or were found in lower abundance in corals from non-oil treatment. It is also worth highlighting the Vibrionaceae family, which was identified only in the oil and in the treatment where the oil came into direct contact with the corals.

Other ASVs found in higher abundance on corals from treatments with oil compared to those from the control treatment, without oil, were the Bacteroidales and Flavobacteriales orders and the *Eubacterium*, *Sediminispirochaeta*, and *Guggenheimella* genera.

The cladogram analysis was conducted to identify the ASVs that exhibited statistically significant differences (*p* < 0.05) in abundance between the control group and each individual oil treatment (1% oil, 2.5% oil, and contact). The number of taxa exhibiting a significant increase was higher in treatments with higher oil concentrations and direct contact. Specifically, the treatments with 1% oil, 2.5% oil, and contact showed significant increases in 4, 8, and 16 ASVs, respectively (Figure 5). Among the taxa that showed an increase in response to the presence of oil, some are opportunistic bacteria potentially pathogenic to corals, such as *Flavobacterium* sp., *Fusibacter* sp., *Sunxiuqinia* sp., *Cohaesibacter* sp., Rhizobiaceae, the Peptostreptococcales–Tissierellales order, and the Clostridia class.

Conversely, the taxa that experienced a significant decrease due to the presence of oil included *Pseudophaeobacter* sp., the Bdellovibrionota phylum, *Dadabacteriales* sp., the Rhodobacteraceae family, and the Stappiaceae phylum.

## 4. Discussion

The experiment conducted with the coral species *M. harttii* and the weathered oil collected during the most extensive oil spill in tropical oceans revealed the PAHs that are still able to solubilize in seawater and their impacts on the coral microbiome, as well as on the photosynthetic efficiency of the Symbiodiniaceae dinoflagellates associated with the coral.

The highest concentrations of the analyzed PAHs in the seawater from treatments with oil were phenanthrene, fluorene, anthracene, pyrene, chrysene, and benzo[a]anthracene. Phenanthrene is very toxic to corals, and Turner et al. [50] observed lethal doses after 48 h in *Acropora cervicornis* up to 33.8% when phenanthrene was at 656 μg L^−1^, and 8.3% mortality at 454 μg L^−1^. Ko et al. [51] analyzed the PAHs that accumulate in coral tissue (*Acropora* sp. and *Montipora* sp.) and noted that phenanthrene, pyrene, and fluorene were the most observed. However, the concentration of PAHs solubilized in water from oil treatments was relatively low when compared to another study that assessed the impact of non-weathered oil on corals [24].

The photosynthetic efficiency of Symbiodiniaceae dinoflagellates associated with *M. harttii* showed an initial drop on days 2 and 3 in all oil treatments, followed by a subsequent drop on day 9 in the treatment where the oil was in direct contact with the coral. Although this drop may be an indication of impact, the Fv/Fm values are still related to a healthy coral [24,52,53]. In a study with the same coral species (*M. harttii*) but with non-weathered oil, the values of Fv/Fm reached close to zero on day 10, which indicates the death of the coral [24]. This observation highlights that highly weathered oil, as employed in the current study, induces substantially less harm to Symbiodiniaceae dinoflagellates compared to non-weathered oil.

Despite the small impact on the photosynthetic capacity of symbiodiniaceans, the weathered oil appears to influence the microbial community associated with the coral. The microbiome of corals was altered in all treatments with oil, mainly in the treatment where the oil was in direct contact with the coral. This alteration in the microbiome can influence the fitness and health of the coral, potentially leading to a state of dysbiosis. Dysbiosis in corals refers to an imbalance or disruption in the composition and function of the coral-associated microbial community, which can negatively affect the health and resilience of the coral host. This may increase the sensitivity of corals to future environmental changes, including those associated with climate change [20].

Some taxa found exclusively or in higher abundance on corals from treatments with oil have been previously reported in earlier studies associated with coral dysbiosis or disease. Vibrionaceae, *Vibrio* sp., Flavobacteriales, and Bacteroidales are indicators of stress in corals [54,55,56]. Bacterial species belonging to the Vibrionaceae family are prominent causative agents of disease in corals [57]. Flavobacteriales appear to be opportunistic pathogens of corals and have been reported in high abundance in white band disease [58]. Bacteroidales also appear to be opportunistic pathogens and have been linked to coral bleaching [56]. 

Cladogram analysis revealed taxa that increased significantly due to the presence of oil. Some are pathogens or indicators of stress in corals. *Flavobacterium* sp. is an opportunistic coral pathogen and has been associated with thermal stress in various coral species [59,60,61,62,63]. *Sunxiuqinia* sp. has been linked to white plague syndrome in *Porites lutea* corals [64]. *Cohaesibacter* sp. is associated with stony coral tissue loss disease (SCTLD) and white plague [65]. *Fusibacter* sp. has been associated with black band disease, white plague disease, yellow band disease, and bleaching in various coral species from different coral reefs [66,67,68]. This genus dominated diseased corals during the SCTLD outbreak, which affected nearly half of the Caribbean coral species [69,70]. Rhizobiaceae is an opportunistic family related to coral disease, including SCTLD [65,69,71]. Peptostreptococcales–Tissierellales have been observed to increase in abundance in bleached coral and may potentially contribute to tissue loss in SCTLD through the production of an alpha-toxin that degrades coral and Symbiodiniaceae dinoflagellates cells [72,73]. Clostridia bacteria have also been associated with SCTLD [72] and have demonstrated an increased abundance in an experiment involving oil and the same coral species investigated in our study, *M. harttii* [24]. Therefore, they have the potential to serve as bioindicators for assessing the impact of oil on *M. harttii* corals.

Among the bacteria that significantly decreased due to the presence of oil stands out *Pseudophaeobacter* sp. and the Bdellovibrionota phylum. *Pseudophaeobacter* sp. is a beneficial bacterium for corals with probiotic characteristics, such as antimicrobial activity against the pathogen *Vibrio* sp., and is found in lower abundance in corals under heat stress [74,75]. Bacteria belonging to the phylum Bdellovibrionota are known to exhibit predatory behavior against pathogens, and certain species are capable of preying upon *Vibrio* sp. [76,77]. Interestingly, in our study, *Vibrio* sp. was exclusively or more abundantly present in corals exposed to oil treatments.

Furthermore, the analysis of the microbiome present in the oil collected from the sea revealed that this highly weathered oil, which has been in the ocean for an extended period, can harbor and transfer bacteria to corals, including potentially pathogenic ones such as Vibrionaceae. This taxon was exclusively detected in the oil samples and in the treatment where direct contact between the oil and the coral was established. Vibrionaceae is a well-known pathogen for coral and other aquatic animals [57]. This raises concerns, as the frequency and intensity of coral disease outbreaks caused by Vibrio pathogens are on the rise [78], posing a significant threat to reef-building corals worldwide. This finding highlights the potential role of oil spills in the dissemination of harmful microorganisms to coral ecosystems. Nevertheless, further research is needed to elucidate the specific interactions between the microbiome of the weathered oil and the coral host, as well as to assess the long-term consequences of bacterial transmission on coral health and ecosystem dynamics.

## 5. Conclusions

The highly weathered oil, stemming from the largest oil spill in tropical oceans, has been demonstrated to exert an impact on the holobiont coral *M. harttii*, primarily through its influence on the coral microbiome. These microbial alterations can lead to dysbiosis in the coral-associated microbial community, increasing the corals’ vulnerability to future environmental changes. In addition, the presence of highly weathered oil, especially in direct contact with corals, stimulates the growth of potentially pathogenic bacteria that can harm corals. Moreover, the oil has been observed to decrease the abundance of potentially beneficial bacteria that play a vital role in maintaining the overall health of corals. Additionally, the oil acts as a carrier for pathogenic bacteria that have effectively colonized corals, potentially facilitating their dispersal and establishment within coral communities. Despite its impact on corals, the overall effect of this weathered oil was relatively lower when compared to non-weathered oils previously studied in coral research. Understanding the specific mechanisms by which the oil affects the coral microbiome can provide valuable insights into the overall impact of the oil spill on coral ecosystems and contribute to the development of effective conservation strategies.

## Figures and Tables

**Figure 1 microorganisms-11-01935-f001:**
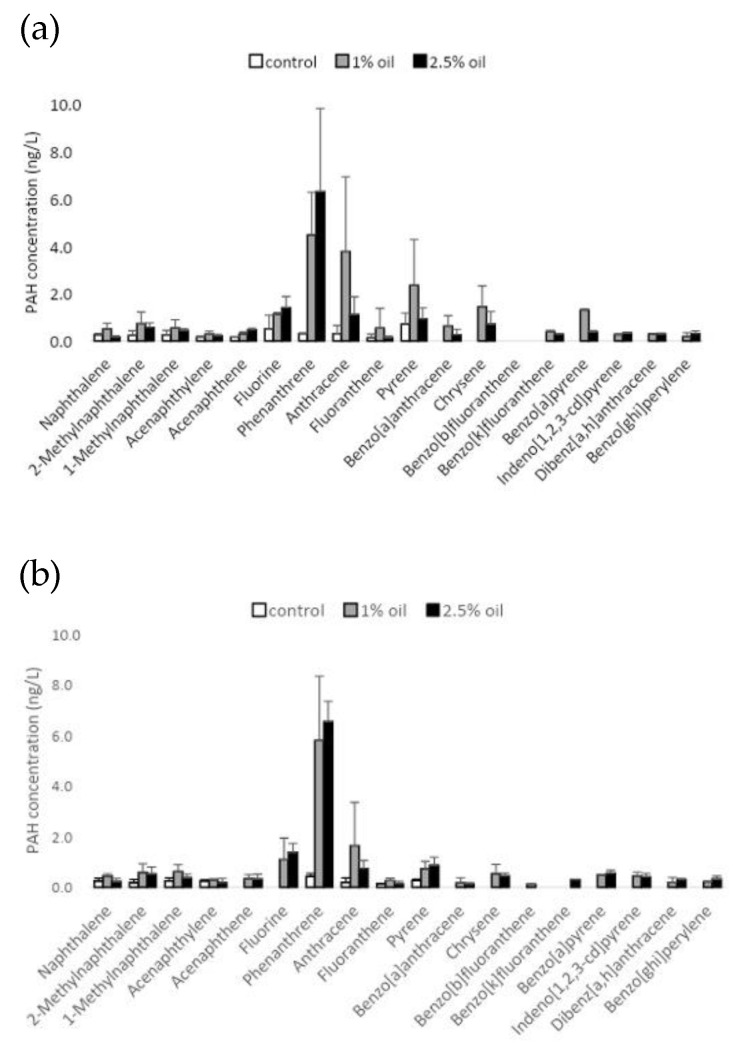
Concentration of polycyclic aromatic hydrocarbons (PAHs) in the water of each treatment (control, 1% oil, 2.5% oil), at 4 days (**a**) and 10 days (**b**) after the exposure of the endemic and endangered coral *Mussimilia harttii* to oil.

**Figure 2 microorganisms-11-01935-f002:**
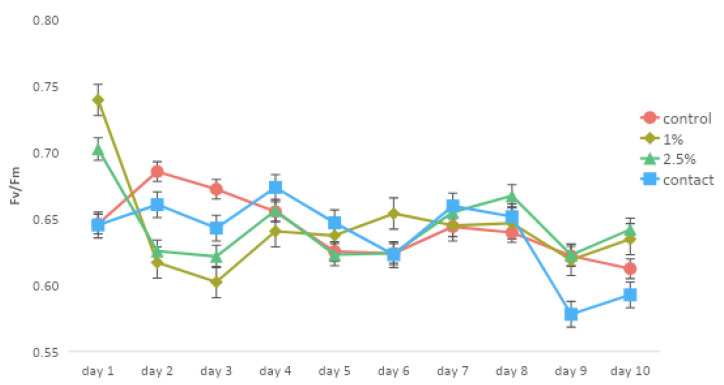
Photochemical efficiencies (Fv/Fm) with mean ± SE of Symbiodiniaceae dinoflagellates from the endemic and endangered coral *Mussimilia harttii* are shown for of the control tank (seawater only), 1% oil, 2.5% oil, and contact with the collected oil.

**Figure 3 microorganisms-11-01935-f003:**
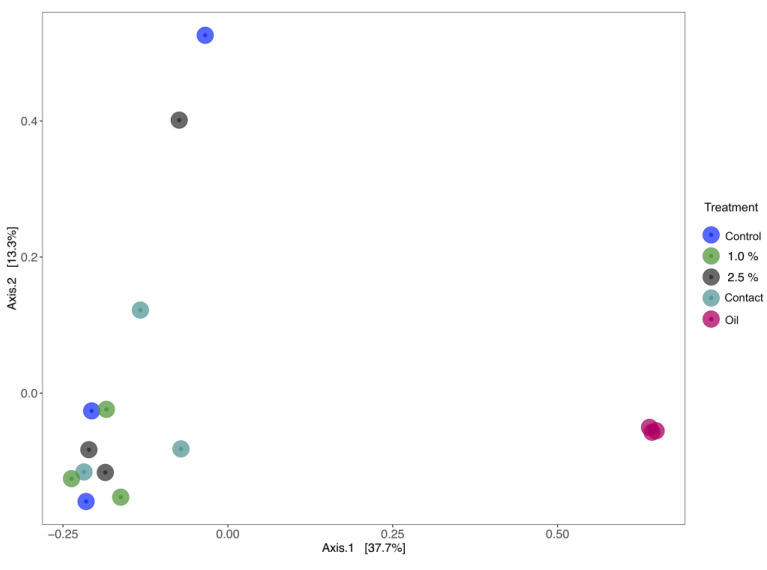
Multidimensional scaling (MDS) of the structure of microbial communities associated with the endemic and endangered coral *Mussismilia harttii*. Treatments: Control, 1% oil, 2.5% oil, contact, and oil.

**Figure 4 microorganisms-11-01935-f004:**
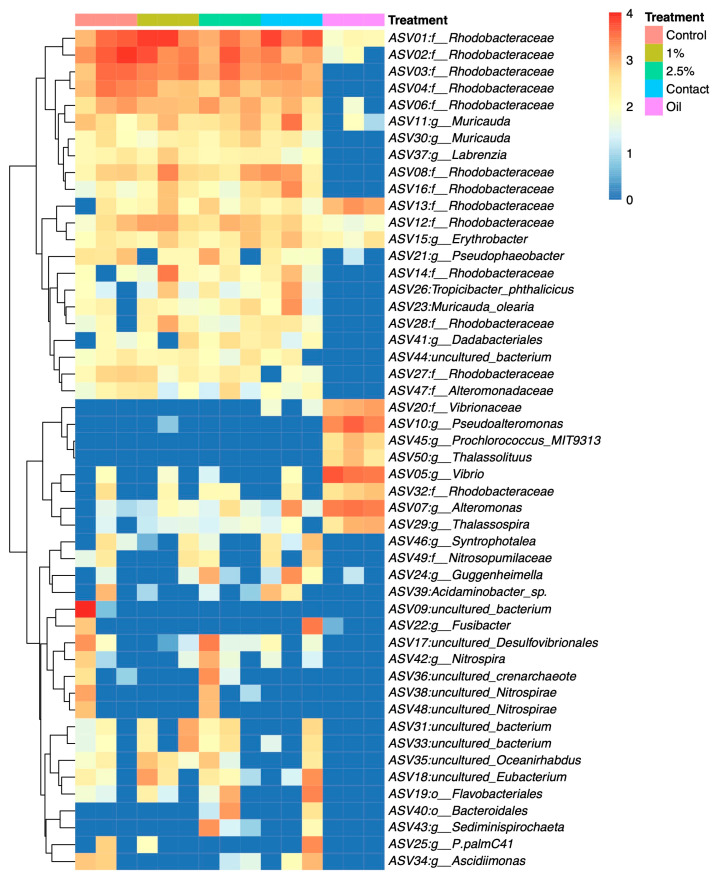
The heatmap of the 50 most abundant ASVs’ relative abundances considering the most abundant taxa in all samples. The bacterial taxa were clustered by the similarity of distribution patterns. Columns separate each of the treatments: Control, 1% oil, 2.5% oil, contact, and oil.

**Figure 5 microorganisms-11-01935-f005:**
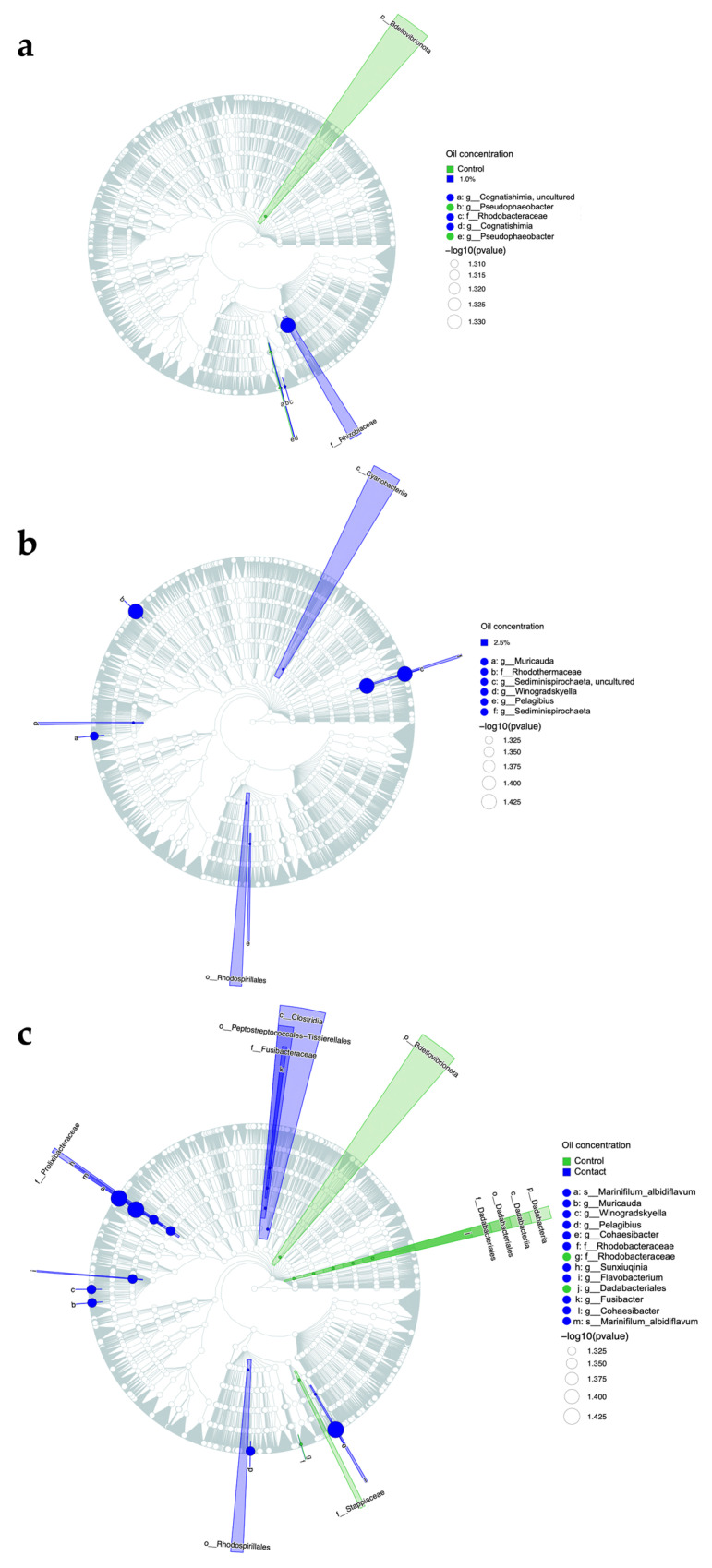
Cladogram representation of coral microbiome taxa associated with the presence or absence of oil. Green indicates taxa enriched in control treatment (without oil), and blue indicates taxa enriched in oil treatments. (**a**) Treatments evaluated were control and 1% oil; (**b**) Treatments evaluated were control and 2.5% oil; (**c**) Treatments evaluated were control and contact. Only taxa with nominal P less than 0.05 are labeled.

## Data Availability

Data are contained within the article.

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
