# Peer review of "The Impact of Highly Weathered Oil from the Most Extensive Oil Spill in Tropical Oceans (Brazil) on the Microbiome of the Coral Mussismilia harttii"

_microorganisms, 2023, doi:10.3390/microorganisms11081935_

Round 1

Reviewer 1 Report

In this study, Pedro et al. analyzed the effect of oil spill on the micriobiome (mainly bacteria) of reef-building Mussismilia harttii living in tropical oceans of Brazil.  They found that the bacterial composition in coral was changed by simulated oil spill conditions in lab scale.  Specifically, they found the relative abundance of potentially pathogenic bacteria increased in oil adding conditions. Several bacteria, significantly decreased due to the presence of oil in cultivation system, included Pseudophaeobacter sp., and Bdellovibrionota phylum. these bacteria were suggested as beneficialmbacterium for corals with probiotic characteristics. they decreasion was possibly related to increase of potentially pathogenic bacteria. 

In all, I think this study is well designated and has environmental indicator values for coral presevation due to oil spill in  specific sea areas. 

 some minor suggestions:

1. The unit of measurement must be standardized throughout the manuscript , such as km2 (line 66), 400 μmol m -2 s -1 (line 110) 

2. Figure 3, for each treatment, were there three individual samples included ? if so, please replace this figure with individual samples, each sample in each treatment or group can used same color or  shapes

3. The resolution of Cladograms (Figure 5) was low, please redrawn this fiture. 

Reviewer 2 Report

Please enter the autor name of the species of corala, and every species that nention în The text for the first time. Check în The text and italized every where you nention name of species. You forgot to italize în one place. Also you write PHA, instead of PAH at Disscusion. 
